# Muscle Ultrasound Echo Intensity and Fiber Type Composition in Young Females

**DOI:** 10.3390/jfmk9020064

**Published:** 2024-04-05

**Authors:** Gerasimos Terzis, Eftychia Vekaki, Constantinos Papadopoulos, Giorgos Papadimas, Angeliki-Nikoletta Stasinaki

**Affiliations:** 1Sports Performance Laboratory, School of Physical Education and Sport Science, National and Kapodistrian University of Athens, 17237 Athens, Greece; eutuxaki@hotmail.com (E.V.); agstasin@phed.uoa.gr (A.-N.S.); 2A’ Neurology Department, Aiginition Hospital, School of Health Sciences, National and Kapodistrian University of Athens, 11528 Athens, Greece; constantinospapadopoulos@yahoo.com (C.P.); gkpapad@yahoo.gr (G.P.)

**Keywords:** ultrasonography, skeletal muscle, muscle strength, muscle quality

## Abstract

Ultrasonography has been extensively used to evaluate skeletal muscle morphology. The echo intensity, i.e., the mean pixel intensity of a specific region of interest in an ultrasound image, may vary among muscles and individuals with several intramuscular parameters presumed to influence it. The purpose of this study was to investigate the correlation between muscle echo intensity and muscle fiber type composition in humans. Thirteen female physical education students (age: 22.3 ± 5.4 years, height: 1.63 ± 0.06 m, body mass: 59.9 ± 7.4 kg) with no history of systematic athletic training participated in the study. Body composition with dual X-ray absorptiometry, leg-press maximum strength (1-RM), echo intensity, and the cross-sectional area (CSA) of the vastus lateralis (VL) muscle according to ultrasonography were measured. Muscle biopsies were harvested from the VL site where the echo intensity was measured. VL echo intensity was not significantly correlated with the percentage of type I muscle fibers or with the percentage area of type I muscle fibers. However, when VL echo intensity was corrected for the subcutaneous fat thickness at the site of the measurement, it was significantly correlated with the percentage of type I muscle fibers (r = 0.801, *p* < 0.01) and the percentage area of type I muscle fibers (r = 0.852, *p* < 0.01). These results suggest that the echo intensity of the vastus lateralis muscle corrected for the subcutaneous fat thickness at the measurement site may provide an estimate of the muscle fiber type composition, at least in young moderately trained females.

## 1. Introduction

Ultrasonography has been extensively used to evaluate skeletal muscle morphology in healthy and diseased individuals, as well as in athletic populations. The echo intensity, which is the mean pixel intensity of a specific region of interest in an ultrasound image, has been increasingly used in the literature as a marker of the quality of skeletal muscles, i.e., muscle strength or power per unit of muscle mass [1,2]. In general, the lower the echo intensity (darker image), the higher the muscle quality is thought to be [3]. Accordingly, in older individuals, higher echo intensities (brighter images) have been linked with increased interstitial fibrous tissue in skeletal muscles, a result that seems to be consistent across studies [4]. Consensus also exists for the link between muscle edema and echo intensity; exercise-induced muscle damage has been shown to increase echo intensity (brighter image) 72 to 96 h after the implementation of a damaging stimulus, and this has been shown to correlate with muscle inflammation [5,6,7]. However, the physiological meaning of muscle echo intensity has been interpreted differently, especially in young exercising populations (for a review, see [8]). For example, investigators have tried to identify a possible link between muscle echo intensity and muscle glycogen concentration [9], but this idea was challenged in a recent study showing that changes in glycogen content do not correlate with changes in echo intensity [10]. Also, changes in muscle strength and echo intensity in response to chronic resistance training are not concurrently observed, which seems to disturb the expected correlation between muscle quality and echo intensity [11,12].

When considering echo intensity, care should be taken for consistency in the measurement procedure. The reliability measures of echo intensity are usually lower than those of other ultrasonography measures, such as muscle thickness, due to technical issues or biological characteristics [8]. One of the biological properties influencing the skeletal muscle echo intensity is the subcutaneous fat thickness at the site of the measurement. It seems that thicker fat layers produce darker ultrasound images, resulting in lower echo intensities, while thinner subcutaneous fat layers produce a lighter ultrasound image, therefore resulting in higher echo intensities [13]. Hence, fat thickness should be accounted for in all measures of echo intensity [14].

Skeletal muscles contain two principal types of contractile cells (fibers), type I and type II, with distinct contractile, metabolic, and structural characteristics. In general, type I fibers are slower with smaller cross-sectional areas and are more resistant to fatigue, while they have wider sarcomere z lines and more capillaries around them than type II muscle fibers do [15,16]. The relative proportions of type I and II muscle fibers fluctuate greatly among different muscles in the same individual; the soleus muscle contains approximately 80% type I fibers, while the long head of the triceps brachii contains approximately 30% type I fibers [17]. Furthermore, there is a large variability in the fiber type composition for the same muscle among individuals. For example, the type I percentage in the vastus lateralis, the most studied human muscle, may vary between 16 and 97% among different individuals [18]. Given the different characteristics of the two types of muscle fibers mentioned above, it would be reasonable to suggest that echo intensity might be different for a certain muscle in individuals with different fiber type compositions. For instance, the higher number of blood vessels around type I fibers might result in a lower echo intensity (darker image) because blood vessels are presented as darker in B-mode muscle ultrasonographs. However, the correlation between the echo intensity and muscle fiber type composition has not been investigated in human muscles. Therefore, the aim of this study was to investigate the relationship between echo intensity and muscle fiber type composition in young, relatively untrained females. It was hypothesized that muscle echo intensity would be correlated with the muscle fiber type composition.

## 2. Materials and Methods

### 2.1. Experimental Approach

Female physical education students participated in this cross-sectional correlational study. Females were recruited because very few data about muscle echo intensity in females exist, and moreover, previous studies from our laboratory showed a relatively large interindividual variability in muscle fiber type composition in the vastus lateralis in young females. Therefore, we hypothesized that a possible correlation between echo intensity and muscle fiber type composition, if it existed, would be clearly revealed in such a population. Participants visited the laboratory three times (Figure 1). On their first visit, anthropometry, body composition assessment, and ultrasonography were performed. On the day after, muscle biopsies were collected from the right vastus lateralis. One week later, the leg press maximum strength (1-RM) was measured. The correlation between the ultrasound vastus lateralis echo intensity and fiber type composition was calculated, among other analyses. The vastus lateralis was chosen for investigation because it is easily accessible for both ultrasonography and muscle biopsy. Body composition and the 1-RM strength were evaluated to identify any possible interference (partial correlations were calculated) with the correlation between echo intensity and fiber type composition.

### 2.2. Participants

Thirteen female physical education students (age: 22.3 ± 5.4 years, height: 1.63 ± 0.06 m, body mass: 59.9 ± 7.4 kg) who were active through university courses but had no experience in systematic resistance training volunteered to participate in the study. The participants were healthy, with no musculoskeletal injuries, and followed a normal diet with no nutritional supplements. They were informed orally and in written form about the research procedures and the possible risks, especially those of the muscle biopsy procedure, and they provided written consent regarding their participation in the study. All procedures were performed in accordance with the principles outlined in the 1975 Declaration of Helsinki, as revised in 2000. All procedures were approved by the Bioethics Committee of the School of Physical Education and Sports Science of the National and Kapodistrian University of Athens (protocol number: 1567/09-10-2023).

### 2.3. Body Composition

The participants were instructed to fast for 12 h and refrain from any strenuous physical activity for 24 h prior to the measurements. Body composition was assessed with dual X-ray absorptiometry (DXA) in the morning hours after the measurement of body height (Seca 213, Surrey, UK) and mass (Tanita BC-545n, Tokyo, Japan). The DXA scan followed 5 min of supine rest on the DXA apparatus (Prodigy Pro, General Electric, Madison, WI, USA). The Lunar Encore v.18 software was used to determine bone mineral density (BMD), body fat mass, and lean body mass (LBM). The intra-class correlation coefficient (ICC) was calculated in a previous setup on two consecutive days (n = 20); for body fat mass, it was 0.99, for the LBM, it was 0.99, and for the BMD, it was 0.99.

### 2.4. Ultrasonography

Ultrasonography was performed after the DXA scans, on the right lower extremity. B-mode ultrasound images (Logiq P9, General Electric, USA) were obtained with a 10–12 MHz linear-array probe for both VL CSA and echo intensity. The participants remained in a supine position for 5 min before starting the procedure. Five additional minutes passed in this position while marking the thigh for imaging (see below) before capturing the first image. For VL CSA, the whole CSA of the quadriceps was imaged to have a clear view of the VL’s boundaries. Specifically, a line was marked from the center of the patella to the medial aspect of the anterior superior iliac spine, and then an axial perpendicular line was drawn at 20 cm of this distance (proximal to the knee). The probe was moved transversely across the thigh on this marked line, taking a continuous single view that pictured the entire CSA of the quadriceps [19]. The CSA of the VL was analyzed using image analysis software (ImageJ v.1.53, U.S. National Institutes of Health, Bethesda, MD, USA). The ICC for VL CSA was previously calculated in our laboratory to be 0.92 (n = 14). Ultrasonography for the VL echo intensity (EI) was performed with a transducer placed perpendicularly to the muscle fascicles with a depth of 40 mm, a gain of 60 dB, and the TGC in the neutral position for all participants. A generous amount of ultrasonography gel was applied between the probe and the skin, with special care being taken to exclude all visible air bubbles and minimize the skin pressure. Before applying the sonography gel, the skin was marked at the point where the acoustic probe was placed to guide the incision for the muscle biopsy. Images were analyzed for echo intensity and for the subcutaneous fat thickness (at the center of the image) with image analysis software (ImageJ, U.S. National Institutes of Health, Bethesda, MD, USA). Subcutaneous fat thickness has a confounding effect on EI; a recent report investigated this effect and provided appropriate correction functions [13]. Because of the different thigh fat thicknesses of the present participants and the fixed focal point, we applied an appropriate equation that was proposed before [13]—EI corrected = EI measured − 5.0054 × F2 + 38.30836 × F (F = subcutaneous fat thickness in mm)—to correct the echo intensity measurements for different subcutaneous fat thicknesses. The ICC for VL echo intensity was calculated to be 0.89 in consecutive days for the vastus lateralis muscle (n = 14).

### 2.5. Muscle Fiber Type Composition

Muscle samples were obtained with Bergström needles from the right lower extremity at the point marked during ultrasonography—under local anesthesia—by a trained medical doctor one day after ultrasonography. Samples were aligned, placed in an embedding compound, and frozen in isopentane, which was precooled to its freezing point. All samples were kept in liquid nitrogen until the day of analysis. Serial cross-sections that were 10 μm thick were cut at −20 °C and stained for myofibrillar ATPase after preincubation at pH 4.3 [20]. Biopsy slices from all subjects were stained at the same time in the same jar. The mean of 298 muscle fibers was classified as type I or II in each sample. The cross-sectional area of all the classified fibers from each sample was measured with image analysis software (ImageJ, U.S. National Institutes of Health, Bethesda, MD, USA) at a known magnification. The percentages of type I and type II muscle fibers were calculated, as was the area of the cross-section covered with type I or type II fibers (i.e., percentage type I fibers × mean CSA of type I fibers). The ICCs in our laboratory for the analysis of type I and type II muscle fiber composition and CSA ranged between 0.93 and 0.96.

### 2.6. 1-RM Strength

One week after the muscle biopsy, the leg press 1-RM of the right lower extremity was evaluated in a 45° leg press apparatus. All participants warmed up with 50 Watts on a stationary bicycle followed by static stretching. Subsequently, they performed 1 set of 10–12 repetitions on the leg press at approximately 40% of the predicted 1-RM, 1 set with 6–8 repetitions at approximately 50–60% of the predicted 1-RM, 1 set with 2–3 repetitions at approximately 65–75% of the predicted 1-RM, and, thereafter, single-repetition sets with increasing loads until they could lift the heaviest load. Three minutes of rest was allowed between sets. All efforts were performed with the right lower extremity only. The highest load was used for the statistical analysis. The ICC for the single-leg press 1-RM strength was calculated earlier in our laboratory (ICC = 0.96, n = 16).

### 2.7. Statistical Analysis

The Shapiro–Wilk test was used to assess the normality, and Lavene’s test was used for the homogeneity of the data. No violations of normality or homogeneity in the distributions were found (*p* > 0.05). Descriptive statistics were used for statistical analysis: mean ± standard deviation. Correlations between variables were examined with the Pearson r correlation coefficient, including the coefficient of determination (Pearson’s r squared, R^2^). The interpretation of the correlations was performed according to Hopkins’ ranking (0.3–0.5 was considered moderate, 0.51–0.70 was considered large, 0.71–0.90 was considered very large, and >0.91 was considered almost perfect) [21]. Partial correlations were also calculated to exclude the effects of muscle mass and strength on the correlations between echo intensity and fiber type composition. Differences between groups were analyzed with the Student *T*-Test. Cohen’s d effect size was also calculated (small: 0.2, medium: 0.5, large: 0.8). The level of significance was set at *p* ≤ 0.05. Reliability for all measurements was assessed using a two-way random-effect intra-class correlation coefficient (ICC). Statistical analysis was performed with JASP software v. 0.18 (University of Amsterdam, The Netherlands).

## 3. Results

The data from the measurements of anthropometry, body composition, muscle morphology, and strength are presented in Table 1. Analysis of the muscle biopsies revealed that six of the participants had >50% type I muscle fibers, and the other seven had >50% type II muscle fibers in their VL (Table 1). Therefore, the data are also presented with the participants assigned in two groups regarding the fiber type composition in the vastus lateralis, i.e., >50% type I muscle fibers and >50% type II muscle fibers (Table 1). Statistical comparisons between these groups revealed that there was a higher subcutaneous fat thickness in the group with a higher percentage of type II fibers. The echo intensity was similar between the groups, but when it was corrected for the subcutaneous fat thickness at the site of the measurement, it was higher in the group with a higher percentage of type II muscle fibers (Table 1).

The echo intensity of the vastus lateralis was not significantly correlated with the percentage of type I muscle fibers, with the percentage area of type I muscle fibers, or with any other of the parameters measured (Table 2, Figure 2). However, the echo intensity of the vastus lateralis corrected for the subcutaneous fat thickness (see the methods for correction) presented a very large correlation with the percentage of type I muscle fibers (r = 0.801, *p* < 0.01, R^2^ = 0.641) and a very large correlation with the percentage area of type I muscle fibers (r = 0.852, *p* < 0.01, R^2^ = 0.725, Table 2, Figure 3). When the effect of 1-RM strength was partialled out, the correlations of the corrected echo intensity with the percentage of type I fibers and percentage of type I fiber area were marginally increased (r = 0.819, and r = 0.870, respectively, *p* < 0.01). When the effect of LBM was partialled out, the correlations of the corrected echo intensity with the percentage of type I fibers and percentage of type I fiber area were also marginally increased (r = 0.85, and r = 0.88, respectively, *p* < 0.01). The echo intensity and the corrected echo intensity were not correlated significantly with any other of the body composition or ultrasonography parameters measured. Also, the echo intensity and the corrected echo intensity were not correlated significantly with the muscle fiber CSA (Table 2).

## 4. Discussion

The aim of this study was to investigate the relationship between echo intensity and muscle fiber type composition in active young females. The main finding was that, when corrected for the subcutaneous fat thickness at the site of the measurement, the echo intensity of the vastus lateralis muscle is highly correlated with the muscle fiber type composition. The fiber type composition largely dictates the contractile and metabolic potential of a muscle [15]. Individuals with a higher percentage of type I muscle fibers in their protagonist muscles are more fatigue-resistant and have a higher potential for aerobic performance, while individuals with a higher percentage of type II fibers are generally more powerful [22]. Here, the echo intensity was similar between the muscles with high and low percentages of type I fibers. However, it seems that the echo intensity is blurred by the subcutaneous fat thickness at the site of the measurement [13]. A thicker fat layer makes the muscle image darker (lower echo intensity). The participants in the present study with a higher percentage of type II fibers in the vastus lateralis had thicker fat tissue at the site of the measurement, resulting in falsely darker images that were of similar echo intensity to the muscles with a higher percentage of type I fibers. When the echo intensity was corrected for the fat tissue thickness, the muscles with a higher percentage of type II fibers presented higher echo intensity values, representing brighter images. So, when the appropriate correction for fat thickness is applied, muscles with a higher percentage of type II fibers are brighter, and muscles with a higher percentage of type I fibers are darker. Interestingly, the calculation of the muscle area covered with type I or II muscle fibers (percentage of CSA of type I or II fibers) shows an even higher correlation with the corrected echo intensity. This might suggest that the echo intensity could be influenced by intra-fiber structures or the connective tissue surrounding the fibers. One possible explanation for this result at the sarcomere level might be that type II muscle fibers have thinner z lines compared to those of type I fibers [16], and this might result in different ultrasound reflections and, therefore, different echo intensities. Also, type I fibers generally have more mitochondria and myoglobin, which might have influenced the echo intensity. However, to the best of our knowledge, no data exist to support these assumptions.

Another explanation for the link between echo intensity and fiber types might be the different capillary supply around type I and II muscle fibers. Type I fibers are usually surrounded by more capillaries than type II fibers in untrained individuals [23]. In general, blood vessels appear darker than muscle tissue in B-mode ultrasonographs. Therefore, muscles with a higher percentage of type I fibers may appear darker in ultrasound images because of the increased number of capillaries. Of note, the number of capillaries in a muscle may change rapidly in response to exercise training [24], unlike the relative proportions of type I and II fibers. Thus, when endurance-trained participants are examined, the correlation between echo intensity and fiber type composition should be lower because of the expected increased capillary density, regardless of the fiber type composition. If this is true, the current data might represent a correlation between echo intensity and muscle capillarity. In fact, the disconnection between fiber type composition and capillary density in the vastus lateralis in trained participants has been shown before [25]. So, the influence of muscle capillarity on echo intensity needs further investigation. Furthermore, type II muscle fibers may be subdivided into type IIa and IIx, with the latter having even fewer capillaries around them, which may contribute to an even brighter ultrasound image. Indeed, the analysis of type IIa and IIx muscle fibers might have provided a better insight into the relationship between echo intensity and fiber type composition, but we were unable to perform this analysis at this point.

### Limitations

This is a correlation study with certain limitations. First, due to the invasive nature of the biopsy procedure, the number of participants was relatively small. However, there was a large variability in the muscle fiber type composition of the participants, which rendered it easier for a correlation with the echo intensity to be revealed. Future studies should include more participants with wider performance backgrounds. Also, the current data were obtained from young females with relatively small subcutaneous fat thicknesses; it is not certain whether these data may be reproduced in males, older individuals, or after a period of systematic resistance training. Regarding exercise training, it has been shown that chronic resistance training may decrease the muscle echo intensity, but these data were reported without a correction for body fat (e.g., [26]). This decrease in echo intensity might be interpreted as a shift towards a slower fiber phenotype (i.e., type II to I) or an increase in the capillarity of the muscle tissue, which is expected with increased resistance training volume. Future studies should address this issue with appropriate corrections for the subcutaneous fat tissue thickness at the site of the echo intensity measurement. Another issue that should be stressed is that the correlation between the echo intensity and the muscle fiber composition would probably not be revealed in older individuals or in individuals with a disease affecting the muscle tissue. In such conditions, the infiltration of muscle tissue by fat and connective tissue usually results in higher echo intensities (brighter images), which, together with a decrease in type II muscle fibers [27], darker images, and thicker subcutaneous fat tissue (darker image), might further blur the relationship between echo intensity and fiber type composition.

## 5. Conclusions

In conclusion, the current data suggest that the echo intensity of the vastus lateralis muscle may provide an estimate of the muscle fiber type composition when corrected for the subcutaneous fat thickness at the site of the measurement, at least in young active females. These results add to the understanding of the interpretation of ultrasound echo intensity in skeletal muscles. Also, the current data, if confirmed with future studies, might suggest that echo intensity may be used as a noninvasive tool to estimate the fiber type composition in the vastus lateralis in a certain group of participants or may assist in allocating these individuals to a type I or a type II muscle fiber group. Future studies should evaluate this relationship in populations of other ages and in other muscles, as well as after long-term exercise training interventions.

### Practical Applications

The current preliminary data suggest that echo intensity might be used to noninvasively estimate the muscle fiber type composition of the vastus lateralis in young females. This finding, if further confirmed in other studies, may have an application in sports talent identification, where the muscle fiber type composition is of importance for sports performance (e.g., sprints, jumps). Moreover, the current data may have relevant significance for certain clinical populations, such as patients with type II diabetes with an altered muscle fiber type composition. The baseline muscle fiber type composition and the possible changes with medical and/or physical exercise treatment in such patients might be revealed with echo intensity measurements. However, this needs further investigation.

## Figures and Tables

**Figure 1 jfmk-09-00064-f001:**
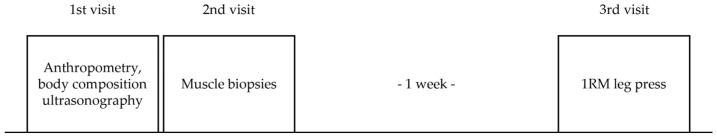
Timeline of the experimental design of the study.

**Figure 2 jfmk-09-00064-f002:**
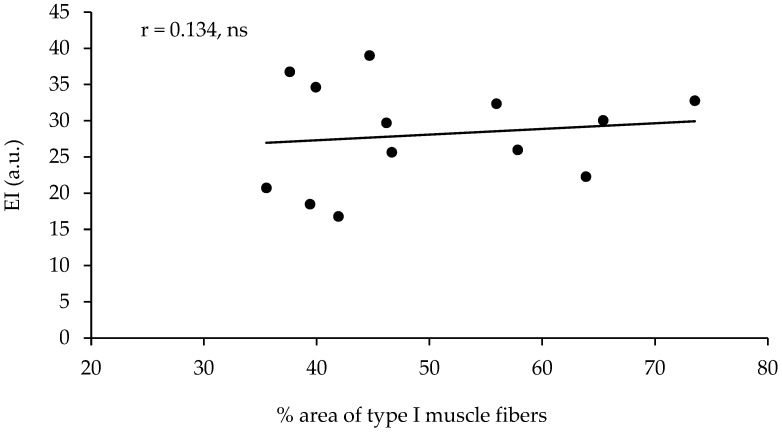
Correlation between the percentage of area covered with type I muscle fibers (percentage of type I CSA) in the vastus lateralis and the echo intensity in thirteen active young females (ns = not significant).

**Figure 3 jfmk-09-00064-f003:**
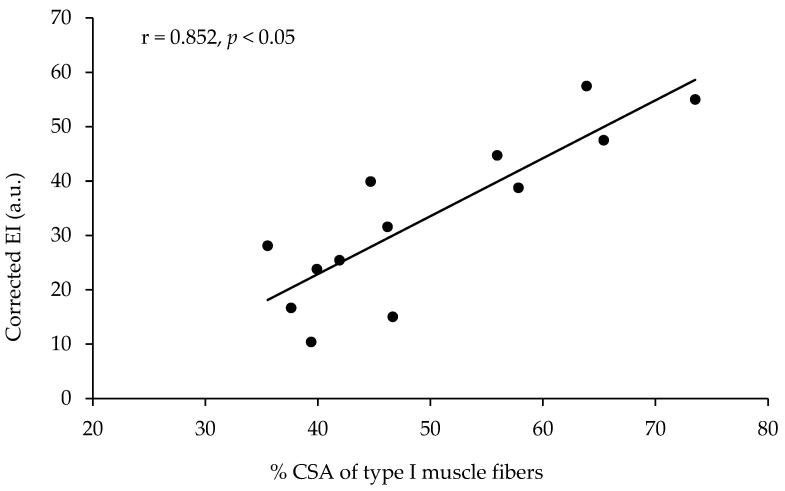
Correlation between the percentage of area covered with type I muscle fibers (percentage of CSA type I) in the vastus lateralis and the echo intensity corrected for fat thickness at the site of the ultrasound measurement in thirteen active young females.

**Table 1 jfmk-09-00064-t001:** Body composition, vastus lateralis muscle morphology, and ultrasonography data in young females (n = 13). Data are also presented for the two subgroups of participants with >50% type I or >50% type II muscle fibers in the vastus lateralis.

	All (n = 13)	Type I > 50% (n = 6)	Type II > 50% (n = 7)	Effect Size Type I–II (Cohen’s d)
**Body composition (DXA)**				
LBM (kg)	41.0 ± 5.2	40.4 ± 6.8	41.6 ± 2.7	−0.234
Fat mass (kg)	16.4 ± 3.3	15.1 ± 2.2	18.0 ± 4.0	−0.886
BMD (g/cm^2^)	1.21 ± 0.10	1.21 ± 0.11	1.20 ± 0.08	0.107
Fat lower limbs (%)	33.0 ± 3.3	31.4 ± 3.5	34.9 ± 1.8	−1.215
**Vastus lateralis morphology**				
Type I fibers (%)	49.4 ± 11.9	40.6 ± 6.1	59.5 ± 8.3 *	−2.623
Type II fibers (%)	50.5 ± 11.9	59.3 ± 6.1	40.4 ± 8.3 *	2.623
Type I CSA (μm^2^)	2780 ± 347	2646 ± 411	2936 ± 179	−0.885
Type IΙ CSA (μm^2^)	2735 ± 487	2556 ± 384	2943 ± 545	−0.883
Type I CSA (%)	49.8 ± 12.1	41.4 ± 4.4	59.7 ± 10.7 *	−2.311
Type IΙ CSA (%)	50.1 ± 12.1	58.5 ± 4.4	40.2 ± 10.7 *	2.311
**Ultrasonography VL**				
VL CSA (cm^2^)	14.8 ± 3.2	13.6 ± 3.5	16.3 ± 2.2	−0.890
Sub/neous fat thickness (cm)	0.61 ± 0.22	0.46 ± 0.15	0.79 ± 0.15 *	−2.131
EI (a.u.)	28.1 ± 7.1	29.2 ± 7.9	26.6 ± 6.2	0.335
Corrected EI (a.u.)	33.4 ± 15.2	23.6 ± 10.3	44.8 ± 11.6 *	−1.927
**Muscle strength**				
1-RM leg press (kg)	109.8 ± 28.3	103.2 ± 37	117.5 ± 10.3	−0.500

VL = vastus lateralis, EI = echo intensity, CSA = cross-sectional area, BMD = bone mineral density, LBM = lean body mass, DXA = dual X-ray absorptiometry. * = statistical significance between the type I and type II groups.

**Table 2 jfmk-09-00064-t002:** Correlation coefficients (Pearson’s r) between the echo intensity (EI) and fiber type composition of the vastus lateralis in young females (n = 13).

	EI (a.u.)	Corrected EI (a.u.)
**Body composition**		
LBM (kg)	−0.220	0.071
Fat mass (kg)	0.069	0.415
BMD (g/cm^2^)	−0.474	−0.118
Fat lower limbs (%)	0.374	0.602 *
**Vastus lateralis morphology**		
Type I fibers (%)	0.076	0.801 **
Type II fibers (%)	−0.076	−0.801 **
Type I CSA (μm^2^)	−0.271	0.156
Type IΙ CSA (μm^2^)	−0.435	−0.100
Type I CSA (%)	0.134	0.852 **
Type IΙ CSA (%)	−0.134	−0.852 **
**Ultrasonography VL**		
VL CSA (cm^2^)	−0.396	0.368
Subcutaneous fat thickness (cm)	0.080	0.984 **
**Muscle strength**		
1-RM leg press (kg)	−0.238	0.183

VL = vastus lateralis, CSA = cross-sectional area, BMD = bone mineral density, LBM = lean body mass. * *p* < 0.05; ** = *p* < 0.01.

## Data Availability

Data is contained within the article.

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
