# Peer review of "Muscle Ultrasound Echo Intensity and Fiber Type Composition in Young Females"

_jfmk, 2024, doi:10.3390/jfmk9020064_

Round 1
Reviewer 1 Report
Comments and Suggestions for Authors
This paper is interesting since it deals with the possibility to assess new evidence about muscles ultrasound and related fibers composition.
Nevertheless, some concerns have to be better addressed.
The introduction is well structured.
The methods section should be drastically improved.
It is necessary to clarify which is the study model. Then, You have to explain why you chose this specific population, that is composed by young females without other specific characteristics. Why not males? why young? Moreover, many sample characteristics are lacking at the baseline. The only common parameter seems to be physical education students, it is not sufficient for obtaining general data and general conclusions. In this sense, how did you previously establish how many persons to be enrolled in the study? A sample size calculation based on a statistical analysis is needed.
Results are clearly presented.
Discussion is really interesting, but in my opinion you should reduce the scope of the results, which do not allow us to draw firm conclusions from this study. Another limit to overcome is the lacking of clinical lapels to your findings, which are interesting even if weak. In order to briefly integrate the discussion in this sense according to the available literature, I suggest the following references:
- Tognolo, L., Coraci, D., Farì, G., Vallenari, V., & Masiero, S. (2022). Validity of ultrasound rectus femoris quantitative assessment: A comparative study between linear and curved array transducers. European journal of translational myology, 32(4), 11040. https://doi.org/10.4081/ejtm.2022.11040
- Chen, K., Hu, S., Liao, R., Yin, S., Huang, Y., & Wang, P. (2024). Application of conventional ultrasound coupled with shear wave elastography in the assessment of muscle strength in patients with type 2 diabetes. Quantitative imaging in medicine and surgery, 14(2), 1716–1728. https://doi.org/10.21037/qims-23-1152
best regards
Author Response
Comments of Reviewer 1.
This paper is interesting since it deals with the possibility to assess new evidence about muscles ultrasound and related fibers composition. Nevertheless, some concerns have to be better addressed.
Response: We wish to thank the reviewer for the time and effort spent on this manuscript as well as for the constructive comments.
_____________________________________________________________________
The introduction is well structured. The methods section should be drastically improved. It is necessary to clarify which is the study model. Then, you have to explain why you chose this specific population, that is composed by young females without other specific characteristics. Why not males? why young? Moreover, many sample characteristics are lacking at the baseline. The only common parameter seems to be physical education students, it is not sufficient for obtaining general data and general conclusions. In this sense, how did you previously establish how many persons to be enrolled in the study? A sample size calculation based on a statistical analysis is needed.
Response: This was a correlational study. Regarding the participants, we recruited young individuals because the correlation between the echo intensity and the muscle fiber composition would probably not be revealed in older individuals or in individuals with a disease affecting the muscle tissue. In these conditions, fat and connective tissue infiltration of muscle tissue, usually results in higher echo intensities (brighter images, Wong et al. 2020) which together with a decrease in type II muscle fibers (darker images) and a thicker fat subcutaneous tissue (darker image) might blur the relationship between echo intensity and fiber type composition. We have addressed this point as a limitation of the study in the discussion. Furthermore, we recruited females because A) very few data exist for muscle echo intensity in females and B) in previous studies from our laboratory young females presented a relatively large interindividual variability in muscle fiber type composition in vastus lateralis, which, we hypothesized would reveal a possible correlation between echo intensity and muscle fiber type composition. These points are now included in the section “Experimental Approach”.
Regarding the characteristics of the participants, we have included data about their anthropometry, body composition (DXA), and muscle strength of the lower extremities. We apologize for not including other health-related parameters such as blood analysis, however, we believe that such data would not add to the study question.
Regarding the sample size population, for a significant Pearson’s correlation of r > 0.6, with a power of 0.6, 12 participants should be included (G*Power 3.1.9.2; 13 participants were finally included). However, please consider that the number of participants in such an invasive study (muscle biopsies) may be limited.
- Wong, V.; Spitz, R.W.; Bell, Z.W.; Viana, R.B.; Chatakondi, R.N.; Abe, T.; Loenneke, J.P. Exercise induced changes in echo intensity within the muscle: a brief review. J Ultrasound, 2020, 23(4):457-472. doi: 10.1007/s40477-019-00424-y.
_____________________________________________________________________
Results are clearly presented. Discussion is really interesting, but in my opinion you should reduce the scope of the results, which do not allow us to draw firm conclusions from this study. Another limit to overcome is the lacking of clinical lapels to your findings, which are interesting even if weak. In order to briefly integrate the discussion in this sense according to the available literature, I suggest the following references:
- Tognolo, L., Coraci, D., Farì, G., Vallenari, V., & Masiero, S. (2022). Validity of ultrasound rectus femoris quantitative assessment: A comparative study between linear and curved array transducers. European journal of translational myology, 32(4), 11040. https://doi.org/10.4081/ejtm.2022.11040
- Chen, K., Hu, S., Liao, R., Yin, S., Huang, Y., & Wang, P. (2024). Application of conventional ultrasound coupled with shear wave elastography in the assessment of muscle strength in patients with type 2 diabetes. Quantitative imaging in medicine and surgery, 14(2), 1716–1728. https://doi.org/10.21037/qims-23-1152
_____________________________________________________________________
Response: We wish to thank the reviewer for this constructive comment. We have included a part in the discussion about the possible clinical applications of the current results.
Reviewer 2 Report
Comments and Suggestions for Authors
MUSCLE ULTRASOUND ECHO INTENSITY AND FIBER TYPE COMPOSITION IN YOUNG FEMALES
General Commentary
This article presents a very interesting and pertinent question of investigate the relationship between the echo intensity and muscle fiber type composition in young, relatively untrained females. It was hypothesized that muscle echo intensity would be correlated with the muscle fiber type composition.
However, some questions need to be clarified in order to better understand and apply the results found.
MODERATE CONSIDERATION
INTRODCTION
Objective and Hypothesis
I suggest that the authors add the determination between fiber type and echo-intensity as an objective. Thus, using r-squared values from linear or non-linear regressions, if applicable. This will strengthen such results found.
MATERIALS AND METHODS
Experimental approach
Please add a timeline figure of the experimental design of the study in this subchapter.
Statistics
Initially, authors must present whether they carried out data normality and homogeneity checks, in addition to which tests were used for this?
Primarily regarding the use of correlations between variables, I suggest that authors, as described in the objectives, use determination models. Using, for example, linear or non-linear regressions that will have case-effect results depending on the case between echo intensity and type of muscle fiber. Furthermore, I suggest that regression classifications be considered for such results.
In the case of correlations, the authors did not classify such results found, please classify and subsequently present such results, as well as discuss them.
In addition to comparing the differences from the T test, I suggest adding the calculation of the effect size and its classification.
RESULTS AND DISCUSSION
After analyzing the statistics with the suggestions presented, please add these findings to the results and, if necessary, to the discussion.
LIMITATIONS AND PRACTICAL OR CLINICAL APPLICATION
Please insert two final sub-chapters of discussion (Limitations and Clinical or Practical Application).
Limitations
What are the limitations of the study, for example in relation to the review carried out, when compared to systematic reviews or meta-analyses, in relation to the description of the studies.
Practical or Clinical Application
What are the applications of the above in this study? Please describe in this chapter (perhaps the last figure should be included here).
CONCLUSION
Insert a sentence about future directions
Author Response
Comments of Reviewer 2.
General Commentary. This article presents a very interesting and pertinent question of investigate the relationship between the echo intensity and muscle fiber type composition in young, relatively untrained females. It was hypothesized that muscle echo intensity would be correlated with the muscle fiber type composition. However, some questions need to be clarified in order to better understand and apply the results found.
Response: We wish to thank the reviewer for the effort spent on this manuscript as well as for the productive comments.
_____________________________________________________________________
INTRODCTION
Objective and Hypothesis
I suggest that the authors add the determination between fiber type and echo-intensity as an objective. Thus, using r-squared values from linear or non-linear regressions, if applicable. This will strengthen such results found.
Response: We wish to thank the reviewer for the comment, and we have added the coefficient of determination for the main results of the study. Although we can understand that this is a positive comment and the reviewer is making an effort to improve this manuscript, we are concerned to discuss these coefficients of determination because here we present initial corelative data and we can only speculate about the physiological mechanisms behind these correlations. For example, we do not feel comfortable stating that approximately 65% of the echo intensity of vastus lateralis may be determined by the muscle fiber type composition of that muscle site, in young females. This is because many other biological attributes could influence ultrasound echo intensity (Wong et al. 2020). Here, we present to the scientific community initial correlation data, and we strongly believe that we have made a good laboratory job obtaining these data. We will continue collecting data on this interesting issue, including long-term training interventions, and if the data support such premise, we will make such statements. Again, we thank the reviewer for this productive comment.
_____________________________________________________________________
MATERIALS AND METHODS
Experimental approach
Please add a timeline figure of the experimental design of the study in this subchapter.to Response: We have now included a timeline figure.
_____________________________________________________________________
Statistics
Initially, authors must present whether they carried out data normality and homogeneity checks, in addition to which tests were used for this?
Primarily regarding the use of correlations between variables, I suggest that authors, as described in the objectives, use determination models. Using, for example, linear or non-linear regressions that will have case-effect results depending on the case between echo intensity and type of muscle fiber. Furthermore, I suggest that regression classifications be considered for such results.
In the case of correlations, the authors did not classify such results found, please classify and subsequently present such results, as well as discuss them.
In addition to comparing the differences from the T test, I suggest adding the calculation of the effect size and its classification.
Response: Normality and homogeneity tests have now been included as well as the coefficients of determination, the classification of the Pearson’s correlations and the effect sizes for the T-tests.
_____________________________________________________________________
RESULTS AND DISCUSSION
After analyzing the statistics with the suggestions presented, please add these findings to the results and, if necessary, to the discussion.
Response: We have now included these analyses in the manuscript.
_____________________________________________________________________
LIMITATIONS AND PRACTICAL OR CLINICAL APPLICATION
Please insert two final sub-chapters of discussion (Limitations and Clinical or Practical Application).
Limitations
What are the limitations of the study, for example in relation to the review carried out, when compared to systematic reviews or meta-analyses, in relation to the description of the studies.
Practical or Clinical Application
What are the applications of the above in this study? Please describe in this chapter (perhaps the last figure should be included here).
Response: We have included these subchapters.
_____________________________________________________________________
CONCLUSION
Insert a sentence about future directions
Response: We wish to thank the reviewer for this comment. We have included a part in the Discussion about possible future experimental directions.
_____________________________________________________________________
Round 2
Reviewer 1 Report
Comments and Suggestions for Authors
The paper seems now better organized.
Considering the intrinsic characteristics and limitations of this study, I think it is now well structured and no further corrections are needed.
Best regards